# Use of Fecal Indices as a Non-Invasive Tool for Nutritional Evaluation in Extensive-Grazing Sheep

**DOI:** 10.3390/ani10010046

**Published:** 2019-12-25

**Authors:** Carla Orellana, Víctor Hugo Parraguez, Wilmer Arana, Juan Escanilla, Carmen Zavaleta, Giorgio Castellaro

**Affiliations:** 1Doctoral Program in Forest, Agriculture and Veterinary Sciences, Campus Sur, University of Chile, Santiago 8820808, Chile; carla.orellanam@gmail.com (C.O.); carmel_ita_1@hotmail.com (C.Z.); 2Faculty of Agricultural Sciences, University of Chile, Santiago 8820808, Chile; vparragu@uchile.cl (V.H.P.); wilmer.arana@gmail.com (W.A.); juanescanillacruzat@gmail.com (J.E.); 3Faculty of Veterinary Sciences, University of Chile, Santiago 8820808, Chile

**Keywords:** annual Mediterranean grassland, fecal indices, sheep, extensive grazing, evaluation, non-invasive tool

## Abstract

**Simple Summary:**

To adequately manage ruminants in extensive grazing, it is important to have non-invasive techniques to evaluate their nutritional status. Some of these techniques are based on the use of fecal indicators, such as the concentration of nitrogen, phosphorus and 2,6-diaminopimelic acid. These indices have been used in some species of wild ungulates, but their use has been limited in domestic ruminants. Hence, this research proposes the evaluation of fecal indices, as predictors of intake of dry matter, nitrogen and phosphorus in sheep that graze on Mediterranean annual grasslands.

**Abstract:**

The aim of the present study was to assess the reliability of fecal indices as predictors of nutrient intake in sheep under extensive grazing conditions. Fecal concentrations of 2,6-diaminopimelic acid (DAPAf), nitrogen (FN) and phosphorous (FP) were determined in four sheep kept in an extensive grazing system on annual Mediterranean grassland in the vegetative, reproductive and dry phenological stages. Metabolizable energy (MEI), crude protein (CPI) and phosphorus (PI) intake were calculated using the botanical composition, metabolizable energy, crude protein and phosphorus concentrations in each vegetal species making up the animal’s diet. Significant differences were observed in the nutrient intake for each phenological stage (*p* < 0.0001). The highest MEI, CPI and PI were observed during the vegetative stage (*p* < 0.0001). FN and FP were different in each phenological stage (*p* < 0.0001), with significant correlations observed between these variables (r = 0.916; *p* < 0.0001). Regressions among nutrient intake and fecal indices were significant, except in the cases of DAPAf and MEI, and DAPAf and CPI. Based on these results, fecal indices could be used to estimate nutrient intake in sheep under extensive grazing on annual Mediterranean grassland.

## 1. Introduction

Fecal indices as predictors of nutrient intake have been widely used to assess the nutritional status in ruminants [1]. These studies have been carried out mostly in wild ungulates and aimed, through non-invasive techniques, to evaluate diet composition, such as protein [2], digestibility of dry matter [3], energy consumption [4] and the intake of minerals such as phosphorus [5] and calcium [6]. Species such as pronghorn (*Antilocapra americana*) [7,8,9], mule deer (*Odocoileus hemionus*) [10,11,12], moose (*Alces alces*) [13], bighorn sheep (*Ovis canadensis*) [14], have shown correlations between fecal nitrogen (FN) and fecal 2,6-diaminopimelic acid (DAPAf) content and diet quality, which can vary significantly according to the environmental conditions [15]. Although FN is a good indicator of protein intake [16,17], its usage is based on the fact that this compound reflects the amount of microbial biomass produced in conditions of high nitrogen availability at the ruminal level. Therefore, higher FN indicates increased microbial protein production [18]. This is similar with respect to the production of DAPAf, since the synthesis of this indigestible compound depends on the proliferation of Gram (-) bacteria in the rumen [19], which would benefit from a rich nutritional diet. In the case of P, the amount found in fecal matter is directly related to its intake [20]. The role of P in the rumen is crucial, especially for cellulolytic bacteria [21], because they depend on P to break down plant cell walls [22]. When the P needs of ruminal microorganisms are not met, microbial activity may be affected [23]. This, in turn, is directly associated with a decrease in dry matter consumption [24]. These fecal compounds have rarely been studied in sheep. It is not known if they change according to the availability and quality of the forage, especially when animals graze on a pasture that directly depends on weather conditions. Therefore, the first objective of the present study was to assess the reliability of fecal indices as predictors of nutrient intake in sheep under extensive grazing conditions. The second objective was to evaluate the effect of different phenological stages of the pasture on the fecal indices, concerning dry matter intake, metabolizable energy, protein and phosphorus.

## 2. Materials and Methods 

The study was conducted in a 1.34 ha paddock of Mediterranean annual grassland, in the Small Ruminants and Dryland Grassland Section of German Greve Silva Experimental Station, Faculty of Agricultural Sciences, University of Chile (33°28’ S, 70°51’ W; 470 m.a.s.l.). It has a semi-arid Mediterranean climate: an average annual temperature of 14.9 °C and an average annual rainfall of 298.9 ± 145.52 mm (±SD), 93.8% of which occurs between April and September [25]. The land belongs to the Durixerolls family [26]. This area has a woody stratum dominated by *Acacia caven* Mol. (“Espino”) and an herbaceous stratum composed of naturalized annual grasses (Genus: *Avena, Aira, Bromus, Hordeum, Vulpia* and *Lolium*) [27,28,29]. In this paddock, four two-year-old merino precoz ewes (average live weight 51.6 ± 8.9 kg at the beginning of the experiment), without pregnancy or lactation, grazed continuously during the vegetative, reproductive and dry phenological stages. The stocking density was about 3.28 dry sheep equivalent/ha. Protocols for animal management and feces sampling were approved by the Animal Care and Research Advisory Committee at the Faculty of Agricultural Sciences, University of Chile. In each phenological stage, a reference herbarium was created with the vegetable species present in the grassland. 

### 2.1. Dry Matter Availability and Grassland Botanical Composition

In each phenological stage, the dry matter availability (DMA, kg/ha) was measured in 10 quadrants of 0.5 m^2^, arranged systematically within the experimental paddock, cutting back the vegetation in the defined area to ground level [30]. In order to obtain the dry matter (DM) content, the fresh collected material was weighed with a digital scale (Kern PCB, Balingen, Germany) and a sub-sample was dried in a forced-air oven (Memmert BP 800, Schwabach, Germany) between 65 and 70 °C, until it reached a constant weight. The DMA was calculated as the average dry matter obtained through the 10 samples (g per 0.5 m^2^), amplified by a factor of 20, in order to express the DM in kg/ha. The grassland botanical composition was evaluated by manually separating and counting plant species found in a sub-sample (50 g) for each quadrant.

### 2.2. Bromatological Analysis of Dry Matter Availability

The dry matter at 105 °C (DM, %), crude protein (CP, %), neutral detergent fiber (NDF, %), acid detergent fiber (ADF, %) and phosphorus (P, %) were determined using the A.O.A.C [31] and Goering and Van Soest [32] methods. The DM digestibility and metabolizable energy (ME) concentration of the pasture were estimated through the ADF using the equations proposed by Horrocks and Vallentine [33] and the Standing Committee on Agriculture (SCA) [34], respectively. This procedure was carried out for each phenological stage.

### 2.3. Feacal Nitrogen (FN), Phosphorus (FP) and 2,6-diaminomipelic Acid Concentrations (DAPAf) Measurements

Fresh feces samples (~5 g), taken directly from the rectum of each animal, were daily obtained during five consecutive days in each grassland phenological stage (vegetative, reproductive and dry). Thus, sample pools from each ewe, representative of each phenological period, were formed (~25 g each; n = 12). The pool samples were dehydrated in a forced-air oven at 65 °C (Memmert BP 800, Schwabach, Germany) until a constant weight was reached, and subsequently ground in a hammer mill (Wiley Mill 3, Philadelphia, PA, USA) with a 1 mm sieve. The FN content (%, organic matter basis; OM) was determined by the Dumas combustion method [35]. In order to determine FP content (%, OM basis), the samples were subjected to acid digestion and subsequently analyzed using a microwave-plasma atomic emission spectrometer in an Agilent Technologies 4200 MP-AES Multi-analyzer (Agilent Technologies, Inc., Santa Clara, CA, USA). The DAPAf (mg g^−1^ OM) was determined using the methodology proposed by Davitt and Nelson [36]. 

### 2.4. Dry Matter Intake (DMI, Kg DM Day^−1^)

The DMI was estimated using the equations proposed by Standing Committee on Agriculture (SCA) [34], factoring the age, the standard reference weight of mature merino precoz sheep (65 kg), the birth weight (data obtained from experimental station records), the actual liveweight (measured in a Tru-test Eziweigh 7 System scale, graduated in kg with precision of 0.5 kg), the grassland DMA and the digestibility of the consumed diet. In order to use these functions, the animals’ live weights were measured.

### 2.5. Diet’s Dry Matter Digestibility (DMS_diet_, %)

Diet’s dry matter digestibility was estimated for each grassland phenological stage through the botanical composition of the diet, which was estimated by the fecal microhistology procedure [37,38]. This technique is based on identifying vegetation epidermis fragments in the feces, which were then compared with the epidermis patterns previously obtained from the identified plant species collected in the study area [39,40,41]. The results of the microhistological analysis were expressed as relative frequency and transformed to density using the Fracker and Brischle (1944) methodology (cited by [39] and [42]). The digestibility of each grassland species present in the diet was estimated based on NDF (%) content using the equation proposed by Horrocks and Vallentine [33]. Estimates of *DMS_diet_* were obtained using the Westoby Equation [43]:(1)$$DMS_{diet}=\overset{n}{\underset{i=1}{\sum }}DMS_{spi}\cdot PROP_{spi}$$
where *DMS_spi_* represents the average digestibility of the species *i* in the diet, *PROP_spi_* the proportion of the species *i* in the diet, and n represents the number of species observed in the fecal sample.

### 2.6. Metabolizable Energy (MEI MJ/day), Crude Protein (CPI, g/day) and Phosphorus (PI, g/day) Intake

We calculated the proportions of metabolizable energy (*ME_diet_*_,_ MJ/kg), protein (*CP_diet_*, g/kg), and phosphorus (*P_diet_*, g/kg) present in the sheep’s diet, using the Westoby equations [43]:(2)$$\;ME_{diet}=\sum_{i=1}^{n}ME_{spi}PROP_{spi}$$
(3)$$CP_{diet}=\sum_{i=1}^{n}CP_{spi}PROP_{spi}$$
(4)$$P_{diet}=\sum_{i=1}^{n}P_{spi}PROP_{spi}$$

In the previous equation, *ME_spi_*, *CP_spi_* and *P_spi_* correspond to *ME*, *PC* and *P* content of the *i* species, respectively. *PROP_spi_* is the proportion of species *i* in the diet. ME concentrations for each species were determined through *DMS_spi_* [34], CP concentration using the Dumas method [36] and P content by atomic emission spectrophotometry. Later, *MEI*, *CPI* and *PI* were calculated using the following equations, where DMI is the dry matter intake:(5)$$MEI=ME_{diet}DMI$$
(6)$$CPI=CP_{diet}DMI$$
(7)$$PI=P_{diet}DMI$$

### 2.7. Statistical Analysis

The results were processed by mathematical and statistical methods using the statistical program STATGRAPHICS Centurion XV. Variables associated with animals (FN, FP, DAPAf, DMI, DMS_diet_, ME_diet_, CP_diet_, P_diet_) were analyzed using a statistical model of repeated means, considering grassland’s phenological stage (vegetative, reproductive, dry) as the main fixed effect, and the animal as a nested effect within each stage. Separation of means attributed to the effect of the grassland’s phenological stages was done using the least significant difference (LSD) test at a significance of 95%, and is presented as means ± standard error of mean. Pearson’s or Spearman’s correlations were calculated between variables, according to data distribution. To determine the magnitude of the association between variables, linear regressions were also calculated [44]. Variables associated to the grassland (DMA, DM, ME, CP, P) were analyzed considering only the effect of their phenological stage.

## 3. Results

### 3.1. Dry Matter Availability and Grassland Botanical Composition, Dry Matter Availability and Bromatological Analysis.

The dry matter availability was different for each phenological stage (*p ≤* 0.0001), and was found to be 60% higher in the dry season than in the vegetative season. The DM_dig_ and ME presented the opposite pattern, decreasing as the pasture matured (*p ≤* 0.0001) (Table 1). The highest concentration of CP was observed during the vegetative period, decreasing by 46% and 64% during the reproductive and dry seasons, respectively (*p ≤* 0.0001). The P content showed a similar trend, decreasing by 46% during the reproductive season and by 54% during the dry season (*p ≤* 0.0001) (Table 1).

The species most frequently found during the grassland vegetative stage were *Erodium moschatum* and *Hordeum murinum*; while during the reproductive stage, *H. murinum*, *E. moschatum* and *Bromus berteianus* were most frequent. During the dry stage, the most abundant species were *H. murinum* and *Erodium spp.* (Table 2).

### 3.2. Nutrient Intake

Dry matter intake was higher during the vegetative and reproductive seasons, decreasing by 4.4% in the dry season (*p* ≤ 0.05). ME_diet_, CP_diet_ and P_diet_ were different between seasons (*p* ≤ 0.05), but all decreased as the grassland dried. ME_diet_ was 16% lower between the vegetative and dry periods. CP_diet_ and P_diet_ decreased by 50% between the vegetative and dry season. MEI, CPI and PI were greater during the vegetative season, whereas they decreased during the reproductive and dry seasons (*p* ≤ 0.05), with a decrease of up to 60% in the case of the CPI (Table 3).

### 3.3. Faecal Nitrogen (FN), Phosphorus (FP) and 2,6-Diaminomipelic Acid Concentrations (DAPAf)

Both FN and FP contents were different in each season (*p* ≤ 0.05). FN had a range of 3.03% to 5.17%, and the highest concentration was observed during the vegetative season. The FP concentration showed a similar pattern to that of FN, with a range of 0.44% to 1.56%, with 1.56% for the vegetative season. This declined by 40% and 73% during the reproductive and dry seasons, respectively. DAPAf concentration showed no differences among seasons (*p* = 0.60) (Figure 1).

Table 4 shows the correlation coefficients for the faecal indices. All indices show a significant level of correlation between them. DAPAf had a positive correlation with FN (*p* ≤ 0.05) and FP (*p* ≤ 0.05); while FN showed a positive correlation with FP (*p* ≤ 0.001).

### 3.4. Relationship between Faecal Indices and Nutrient Intake

Dry matter intake showed a significant correlation with the three faecal indices, mostly with FP (*p* ≤ 0.001). The regression between MEI and DAPAf showed a linear trend, although this was not significant (*p* = 0.0518), whereas the regressions between MEI–FN and MEI–FP were significant (*p* < 0.001; R^2^ = 0.92). The regression CPI–DAPA also showed a linear trend, although this was not significant (*p* = 0.0746). However, the regressions CPI–FN and CPI–FP were significant (*p* < 0.001; R^2^ = 0.9 and 0.8, respectively). The regressions between PI and all faecal indices were significant (*p* < 0.005; R^2^ = 0.4 for DAPAf and 0.8 and 0.9 for FN and FP, respectively) (Table 5).

## 4. Discussion

Results of the present study indicate that, in grazing sheep, FN and FP concentrations may properly estimate the nutrient intake. Therefore, fecal indicators and predictive equations could be employed as a non-invasive nutritional assessment tool, as well as a research tool, through the collection of fresh feces and determination of nitrogen and phosphorus. However, it is suggested to validate these equations in other pastoral conditions, taking into account different grassland types and, if possible, a greater number of animals.

### 4.1. Grassland Dry Matter Availability and Nutrient Concentration

In the grassland, DMA observed during the seasons under study was similar to that reported by Castellaro and Squella [45] for Mediterranean grassland. These authors have observed from 1000 kg/ha at the beginning of the vegetative season up to 4000 kg/ha at the end of the dry season. The maximum DMA identified in our study was 3550 kg/ha during the dry season, an amount closer to the average indicated by Acuña et al. [46] for a pasture with similar characteristics (3035 kg/ha). Therefore, the DMA levels measured in the grassland would not be affecting the ewe’s voluntary dry matter intake. A decrease of DM digestibility during the dry season was also observed by Olivares [29], but was linked to in vitro organic dry matter digestibility, with values of 56.9% and 53.0% for spring and summer, respectively. In addition, Soto [47] indicated changes in digestible energy from 11.97 to 11.05 MJ/kg OM for spring and summer, respectively. The ME values observed during this study were similar to those reported by George and Bell [48] and George [49] for annual grassland in California, which had characteristics similar to those in our experiment. The high concentrations of CP in the grassland can be accounted for by the nitrogen uptake capacity of the grasses. In grasses, the accumulation of N is directly associated with the soil N content, which would be toxic even under grazing conditions [50]. There are few reports regarding P concentration in Mediterranean annual grassland. However, the decrease of this mineral towards the dry period has been reported by George [49], in annual California grasslands, with levels similar to those found in this trial, e.g., 0.45% to 0.20% between the vegetative and dry periods, respectively. DMA, EM, PC and P levels decreased as the pasture dried. This situation corresponds to the normal process of growth and development in plant species, where during the progression of phenological stages, cellular content decreases, then energy concentration, protein, vitamins and minerals also decrease [47,48,49,50,51,52,53].

### 4.2. Dry Matter, Metabolizable Energy, Crude Protein and Phosphorus Intake in Sheep.

Nutrient intake presented a similar pattern to that of the grassland nutrient concentration, decreasing as the grass dried. Environmental changes directly affect nutrient intake in grazing animals, possibly forcing them to use a more efficient grazing strategy. Daily DMI was always higher than 1.1 kg, seemingly indicating that the animals satisfied their nutritional requirements [54], even during the dry period. The decrease in DMI towards the end of the trial could be related to the increase of fiber content in grassland plant species, thereby reducing diet digestibility from 71% in the vegetative period to 61% in the dry period. The higher dietary fiber content takes up greater ruminal volume and has a greater retention time in the digestive tract, leading to lower intake [55]. The MEI, CPI and PI of the animals also met the sheep’s nutritional requirements during the three phenological stages of the prairies. According to Court et al. [56], the ME requirements for adult sheep (60–65 kg) in the maintenance stage are 8.9 MJ/day, thus, even during the dry period, the intake of ME was enough to meet the sheep’s nutritional demands. Regarding protein requirements, NRC [54] considers a demand equivalent to 113.3 g/day of CP during the maintenance stage for adult sheep weighing 70 kg, a requirement that is less than the amount of CP consumed during the dry period. Likewise, the CP also satisfied the sheep’s requirements during all of the periods, according to the NRC [54].

### 4.3. Faecal Nitrogen (FN), Phosphorus (FP) and 2,6-Diaminomipelic Acid Concentrations (DAPAf)

Fecal indices reflected changes in nutrient intake. Likewise, fecal indices indicate changes in the ruminal metabolism, which would be at a maximum during the vegetative stage. During this stage, nutrient intake would be higher in order to promote microbial development, allowing the consequent increased availability of metabolites to be absorbed by the rumen walls. The three fecal indices studied showed a decrease in concentration as the grassland dried, especially in the cases of FN and FP. Similar variations in fecal indices have also been observed in some species of wild ruminants [57], e.g., DAPAf and FN in pronghorn (*Antilocapra americana*) [7] and in bighorn sheep (*Ovis canadensis*) [14]. There are no recent studies on sheep DAPAf concentration. However, the range obtained in this study for dry matter (0.49–0.59 mg/g), was higher than those observed by McKinney et al. [14] in bighorn sheep, 0.23 to 0.56 mg/g. The previous behavior could be attributed to metabolic differences in microbial activity between domestic and wild sheep. FN content presented a variation between 3.0% to 5.2% (OM base), which were higher than those specified by Peripolli et al. [58], who reported values from 0.96% to 4.9% in sheep consuming Rhodes grass hay (*Choris gayana* Kunth) and rye hay (*Secale cereale* L.), respectively. Wang et al. [3], reported concentrations of FN, ranging from 1.2% to 4.3%. Additionally, Escanilla [59] indicated a range of 1.14% for the dry period to 2.69% in the vegetative period, for a location close to that of the present experiment. The differences between our results and those of other researchers could be due to the botanical composition of the studied areas, and therefore, n would be affecting then metabolism. For instance, it is possible that a higher grass content in the grassland increases CP consumption in sheep, which would eventually result in a higher concentration of FN [60]. Regarding the FP observed (range 0.4% to 1.5%), it was lower than that reported by Mellado et al. [61] for sheep, ranging from 1.2% to 2.1% when they grazed on a desert grassland in Northern Mexico, during the summer and spring, respectively. The variations observed in the indices analyzed could be closely related to the changes in grassland nutrient concentrations [8], considering that the fecal indicators reflect their diets [15]. In the case of DAPAf, several authors indicate that its presence in the feces is associated to grassland dry matter digestibility and consequently, with the available energy at the ruminal level for microbial production [4,60,62]. Higher ruminal bacteria content produced by high grassland energy concentrations would be reflected through DAPAf, which is an indigestible compound [8,63]. The increase in FN content during the vegetative period and its marked decrease during the dry period would be a reflection of the changes in grassland CP content, which in itself is associated to its phenological stage [60]. The protein found in the grassland is the main source of nitrogen for ruminal bacteria, so greater protein availability associated with greater dry matter digestibility, would encourage bacterial growth at the ruminal level [34,62,64]. Crude protein concentration and DM_dig_ were higher in the grassland during the vegetative period, which would explain the high FN content during that period. The decrease of FP content towards the dry period would be a result of lower plant phosphorus concentrations during each phenological stage, as reported by Rowarth et al. [65]. Although the mean P values for each phenological stage were higher than those reported by McDowell et al. [66], it was possible to observe a decrease in its concentration as the pasture matured. According to McDowell et al. [66], the P concentration of plants can be 0.3% in early spring, decreasing to 0.15% when they mature. Fecal P loss, P absorption, retention, urinary excretion and salivary recycling are directly related to P intake as part of the organism’s homeostatic mechanism [20], which would explain the changes in the FP that were observed. 

Several studies have used fecal indices as nutritional predictors, most of them for wild animals [67]. In particular, DAPA has been used as an indicator of microbial protein, although this was for duodenal content samples and not from fecal matter [68]. There are few studies that have used DAPAf in sheep, however in other species of ruminants such as elk (*Cervus canadensis*) [13] and pronghorn (*Antilocapra americana*) [7], clear variations in the annual profile of DAPAf concentration in extensive grazing conditions have been observed. For this study, it is possible that the lack of a significant decrease for this indicator in fecal samples resulted probably from the individual variation of DAPAf concentrations in each phenological stage, which was also reported by McDonald [9] in pronghorn. This aspect also suggests that in future studies, of observations should be used. FN is the most important and accurate indicator of nutrient intake in ruminants [15]. There are several studies that have used FN to estimate CP consumption and DM digestibility [7,8,14]. Semebia [1], in addition, mentioned that FN is the indicator that has the greatest potential to be used to estimate DM intake levels, diet digestibility and diet CP content. The seasonal variation of FN has even been observed in roe deer diets (*Capreolus capreolus*), where secondary compounds such as tannins would modify the effectiveness of FN as an indicator of protein intake. However, FN can properly reflect the changes in diet quality throughout the year [67]. These examples explain the variations of FN observed in sheep during the three phenological stages. The FP, as an efficient marker of the P intake in sheep, was also observed by Mellado et al. [61], with seasonal variations from 0.8% in autumn to 2.1% in spring, on dry matter basis. Studies in other ruminants such as springbok (*Antidorcas marsupialis*), blue wildebeest (*Connochaetes taurinus*) and goats (*Capra aegagrus hircus*), have also shown substantial changes in FP concentration according to the dry or rainy season in semi-arid grasslands [69,70]. The above equations to estimate nutrient intake in extensive grazing-sheep should be validated in different grassland types, since the type and number of consumed plants and the changes in nutrient concentration that they experience throughout their life cycle could affect these relationships. 

### 4.4. Correlations among DAPAf, FN and FP

Although each fecal index has been associated with a specific component of the diet, there are certainly interactions among all dietary nutrients (therefore, among fecal indices), which influence the rumen fermentative capacity, causing changes in the microbial mass and in the quantity of volatile fatty acids available for the ruminant [71]. Consequently, fecal indices and animal nutritional status are closely related. The interaction between the three fecal indicators was also observed in mule deer (*Odocoileus hemionus eremicus*) by Carrera et al. [12] and in bighorn sheep (*Ovis canadensis*) by Mckinney et al. [14]. Both authors showed positive and significant correlations between FN and DAPAf. Mellado et al. [61], on the other hand, indicate that the FN and the FP would be strongly related in sheep, which is consistent with the results reported in this study. It is difficult to analyze each individual indicator separately, due to the complexity of the ruminal system [71] and the complex interactions among microorganisms that are found there. An example of this would be the high correlation found between FP and CPI, as well as the degree of adjustment for the regression equation between both variables; and the estimated positive association between FP and dry matter intake. In this regard, it has been demonstrated that P-deficient diets could lead to a decrease in voluntary food intake [72], which consequently leads to a decrease in energy and protein intake.

## 5. Conclusions

Fecal nitrogen and phosphorous indices, but not DAPAf, vary depending on the phenological stage of the annual Mediterranean grassland, reflecting the nutrient concentration in accordance with its botanical composition and grassland phenological stage. The FN and especially, the FP content, can be used as predictors of DMI, ME, CP and P in sheep under extensive grazing systems on annual Mediterranean grassland, with an adequate level of precision. These indicators can be used as an efficient non-invasive method to adjust sheep nutritional management. However, it is suggested to evaluate these indices in other pastoral situations, using a larger number of animals.

## Figures and Tables

**Figure 1 animals-10-00046-f001:**
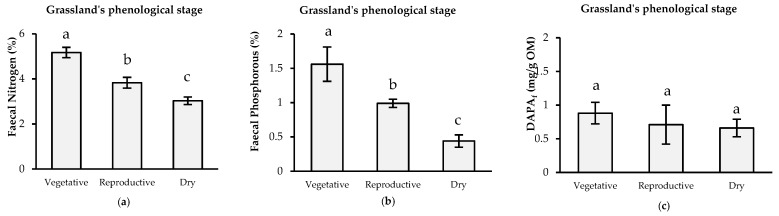
(**a**) Faecal nitrogen (FN), (**b**) phosphorus (FP) and (**c**) 2,6-diaminomipelic acid (DAPAf), concentrations in each grassland’s phenological stages at the experimental site. ^a,b,c^ Different letters indicate a statistically significant difference between periods according to least significant difference (LSD) Test (*p* ≤ 0.05). Bars around the mean indicate the standard desviation.

**Table 1 animals-10-00046-t001:** Dry matter availability (DMA), digestibility (DM), metabolizable energy (ME), crude protein (CP) and phosphorus (P) concentration in the grassland, at different phenological stages.

Component	Grassland’s Phenological Stage			SEM ^1^	*p* Value
	Vegetative	Reproductive	Dry		
DMA (kg/ha)	1361.9 ^c^	2173.1 ^b^	3549.6 ^a^	226.47	<0.0001
DM (%)	69.20 ^a^	63.50 ^b^	57.56 ^c^	0.91	<0.0001
ME (MJ/kg)	10.20 ^a^	9.22 ^b^	8.19 ^c^	0.16	<0.0001
CP (%)	29.15 ^a^	15.77 ^b^	10.39 ^c^	1.42	<0.0001
P (%)	0.54 ^a^	0.37 ^b^	0.23 ^c^	0.03	<0.0001

^a,b,c^ Means bearing different superscripts in the same row are significantly different according to the LSD test (*p* ≤ 0.05).^1^ SEM, standard error of mean (n = 30).

**Table 2 animals-10-00046-t002:** Botanical composition of grassland in each phenological stage in the experimental grassland (%, mean ^1^ ± standard desviation).

Species	Grassland’s Phenological Stage		
	Vegetative	Reproductive	Dry
**Perennials grasses**			
*Phalaris aquatica L.*	nd	nd	0.6 ± 2.0
*Thinopyrum ponticum (Podp.) Z.W. Liu and R. R. -C. Wang*	nd	nd	2.9 ± 9.2
**Annual grasses**			
*Bromus berterianus Colla.*	4.1 ± 13.1	18.4 ± 18.5	10.8 ± 18.0
*Bromus hordeaceus L.*	nd	5.8 ± 17.3	5.9 ± 18.6
*Hordeum murinum L. subsp. leporinum (Link) Arcang.*	41.6 ± 25.0	18.5 ± 20.0	33.2 ± 32.2
*Phalaris minor Retz*	nd	2.6 ± 6.2	0.3 ± 0.6
*Vulpia sp.*	9.0 ± 15.1	5.7 ± 7.6	11.4 ± 15.7
*Lolium multiflorum Lam.*	nd	nd	0.5 ± 1.4
*Poa annua L.*	4.4 ± 13.3	0.4 ± 1.1	0.06 ± 0.09
*Briza minor L.*	nd	0.02 ± 0.1	
**Total grasses**	59.1 ± 24.4	51.6 ± 19.2	65.6 ± 25.7
**Graminoids**			
*Juncus bufonius L.*	nd	0.9 ± 2.9	0.2 ± 0.4
**Total graminoids**	nd	0.9 ± 2.9	0.2 ± 0.4
**Annual herbs**			
*Amsinckia menziessi (Lehm.) A. Nelson and J.F. Macbr.*	4.9 ± 8.4	2.1 ± 5.6	1.4 ± 2.4
*Anthemis cotula L.*	0.2 ± 0.6	1.3 ± 3.5	0.6 ± 1.1
*Brassica rapa L.*	nd	2.5 ± 5.3	0.7 ± 2.3
*Chenopodium album L.*	nd	0.03 ± 0.1	1.8 ± 5.6
*Chenopodium murale L.*	nd	nd	2.7 ± 8.4
*Capsella bursa pastoris L.*	2.6 ± 4.1	2.2 ± 2.2	1.1 ± 1.6
*Erodium spp ^2^.*	nd	nd	24.0 ± 27.0
*Erodium botrys (Cav.) Bert.*	0.5 ± 1.2	1.0 ± 2.8	nd
*Erodium cicutarium (L.) L’Hér.*	nd	3.2 ± 9.5	nd
*Erodium moschatum (L.) L´Hér.*	29.8 ± 22.4	33.8 ± 23.5	nd
*Hypochaeris sp.*	0.6 ± 1.4	nd	nd
*Malva parviflora L.*	1.2 ± 3.3	nd	0.1 ± 0.2
*Medicago polymorpha L.*	0.02 ± 0.04	0.8 ± 2.0	0.6 ± 1.8
*Plagiobothrys procumbens (Colla) Colla.*	nd	0.5 ± 1.2	nd
*Sisymbrium irio L.*	0.1 ± 0.2	nd	nd
*Sonchus sp.*	nd	nd	1.3 ± 3.6
*Urtica urens L.*	0.9 ± 2.8	0.01 ± 0.04	nd
**Other annual herbs**	0.1 ± 0.2	0.3 ± 1.0	nd
**Total annual herbs**	41.0 ± 24.4	47.6 ± 21.3	34.2 ± 25.9

nd: Species not detected in the sample. ^1^ Refers to the average obtained in 10 evaluation quadrants of 0.25 m^2^. ^2^ Mainly *E. moschatum.*

**Table 3 animals-10-00046-t003:** Effect of annual grassland phenological stages on dry matter and nutrient intake in grazing sheep.

Component ^1^	Grassland’s Phenological Stage			SEM ^2^	*p* Value
	Vegetative	Reproductive	Dry		
DMI (kg/day)	1.60 ^a^	1.56 ^a^	1.51 ^b^	0.01	0.0123
DMS_diet_ (%)	70.80 ^a^	64.10 ^b^	60.80 ^c^	0.28	<0.0001
ME_diet_ (MJ/kg)	10.47 ^a^	9.32 ^b^	8.76 ^c^	0.22	<0.0001
CP_diet_ (g/kg)	324.24 ^a^	187.76 ^b^	134.78 ^c^	2.41	<0.0001
P_diet_ (g/kg)	4.43 ^a^	3.73 ^b^	1.98 ^c^	0.03	<0.0001
MEI (MJ/day)	16.78 ^a^	14.56 ^b^	13.18 ^c^	0.46	<0.0001
CPI (g/day)	519.75 ^a^	293.38 ^b^	202.98 ^c^	40.31	<0.0001
PI (g/day)	7.10 ^a^	5.82 ^b^	2.98 ^c^	0.52	<0.0001

^1^ DMI, dry matter intake; DMS_diet_, diet’s dry matter digestibility; ME_diet_, diet´s metabolizable energy concentration; CP_diet_, diet’s crude protein; P_diet_, diet’s phosphorus concentration; MEI, metabolizable energy intake; CPI, crude protein intake; PI, phosphorus intake. ^a,b,c^ Means bearing different superscripts in the same row, significant difference according to the LSD test (*p* ≤ 0.05). ^2^ SEM, standard error of mean (n = 12).

**Table 4 animals-10-00046-t004:** Pearson correlation coefficient (*r, p*-values) between faecal indices.

Faecal Indices	DAPAf	FN	FP
DAPAf	---	r = 0.608, *p* ≤ 0.05, n = 12	r = 0.610, *p* ≤ 0.05, n = 12
FN	r = 0.608, *p* ≤ 0.05, n = 12	---	r *=* 0.916, *p* ≤ 0.01, n = 12
FP	r = 0.610, *p* ≤ 0.05, n = 12	r = 0.916, *p* ≤ 0.01, n = 12	---

Faecal 2,6-diaminomipelic acid (DAPAf, mg g^−1^ OM basis), faecal nitrogen (FN, % OM basis) and faecal phosphorous (FP, %, OM basis).

**Table 5 animals-10-00046-t005:** Relation between nutrient intake (Y, g day^−1^) and fecal indices (X, mg g^−1^ OM or %) in grazing sheep.

Intake ^1^	Fecal Index ^2^	Regression Equation	n	R^2^	SEM ^3^	*p* Value
DMI	DAPA_f_	Y = 1.382 + 0.225X	11	0.481	0.039	0.0179
	FN	Y = 1.387 + 0.043X	12	0.635	0.032	0.0019
	FP	Y = 1.468 + 0.089X	12	0.775	0.025	0.0002
MEI	DAPA_f_	Y = 10.051 + 6.147X	11	0.358	1.390	0.0518
	FN	Y = 8.379 + 1.613X	12	0.927	0.447	<0.0001
	FP	Y = 11.815 + 3.040X	12	0.917	0.478	<0.0001
CPI	DAPA_f_	Y = −54.702 + 505.641X	11	0.311	127.091	0.0746
	FN	Y = −237.626 + 143.784X	12	0.943	34.953	<0.0001
	FP	Y = 76.742 + 262.935X	12	0.878	51.148	<0.0001
PI	DAPA_f_	Y = −0.844 + 7.764X	11	0.431	1.505	0.0281
	FN	Y = −1.646 + 1.733X	12	0.817	0.811	0.0001
	FP	Y = 1.848 + 3.467X	12	0.910	0.568	<0.0001

^1^ DMI, dry matter intake (kg/day); MEI, metabolizable energy intake (MJ/day); CPI, crude protein intake (g/day); PI, phosphorus intake (g/day). ^2^ DAPA_f_, 2,6-diaminomipelic acid fecal (mg/g MO); FN, fecal nitrogen (%); FP, fecal phosphorous (%). ^3^ SEM, standard error of mean (n = 12).

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
