# Peer review of "Use of Fecal Indices as a Non-Invasive Tool for Nutritional Evaluation in Extensive-Grazing Sheep"

_animals, 2019, doi:10.3390/ani10010046_

Round 1

Reviewer 1 Report

General Comments

Authors have a novel idea in this study which should have wider research applications in future especially in the extensive grazing systems all over the world. It is a well-researched and nicely described study.  Introduction needs more details on the overall framework of the study and significance of the study in context of its rationale, contribution to the field and practical implications. The methodology is particularly well described. Presentation of results can be enhanced by graphical representations of the association between variable instead of the table of correlation indices. Discussion is good in general, but a few subsections needs more vigorous discussion on the results of this study rather than simply comparing the results with previous studies (I have pointed out at appropriate places). Some lines should be written on the limitations of the study, implications of the study and more details on the direction of future research strategies on these indices.

I congratulate the authors for a thorough research work and a professionally written manuscript.

Specific Comments

Simple summary

Line 18: Replace ‘annual ranges’ with ‘grasslands’

Abstract

Line 19: Replace ‘capacity’ with ‘reliability’

Line 23: Delete ‘intake’.

Line 31: Replace ‘indexes’ with ‘indices’

Keywords: I suggest the following as key words: Mediterranean, grassland, faecal indices, sheep, extensive grazing, evaluation, non-invasive tool

Introduction

Line 36:  Replace ‘indexes’ with ‘indices’ here and all through the manuscript.

Line 37: Please check this word ‘seek’. I don’t understand what it means.

Line 41: Delete ‘among others,’

Line 43: Replace ‘depending on’ with ‘according to’

Line 46: Please confirm this ‘Therefore, higher FN leads to increased microbial protein production’. I think it should be ‘Higher FN indicates increased microbial protein production’

Line 46-47: Reframe ‘Something similar occurs in the case of..’ as ‘This is similar to the production of …’

Line 48: Replace ‘highly’ with ‘rich’

Line 49-50: Reword ‘….. the intake of such a mineral….’ As ‘… to its intake’.

Materials and methods

Lines 63-65: Reframe ‘It has a semi-arid Mediterranean climate (average annual temperature of 14.9 °C), with an average annual rainfall of 298.9 ± 145.52 mm (±SD), with 93.8% falling between April and September [25]. As ‘It has a semi-arid Mediterranean climate: average annual temperature of 14.9 °C, average annual rainfall of 64 298.9 ± 145.52 mm (±SD), 93.8% of which occurs between April and September [25].’

Line 82: Rewrite ‘……. factor of x20’ as ‘… factor of 20’

Line 137-38: I suggest more explicit explanation for the readers for the ‘Linear regression equations were calculated in order to determine the mathematical associations between variables’ as ‘The Generalized linear model (GLM) was used to determine the mathematical associations between the variables’

Results

Line 150: Table 1 – Replace ‘…, in different phenological stages’ with ‘…., at different phenological stages

Discussion

4.1 Grassland dry matter availability and nutrient concentration – The discussion is solely based on comparing your results with other studies. I would like more reasoning/discussion on your results.

Line 207: Reword ‘…employed for non-invasive nutritional assessment, as well as a research tool…’ as ‘…. employed as a non-invasive nutritional assessment tool…’

Line 211: Insert ’DMA’ in ‘…observed DMA from…’

Line 223: Replace ‘antecedents’ with ‘reports’

Lines 280-81: Please explain what happens to Phosphorus concentrations, increase or decrease.

Line 283: Reframe ‘it was possible to observe a decrease in this mineral as the pasture matured’ as ‘ ‘a decrease in its concentration is possible as the pasture matures’

Line 288: Replace ‘estimators’ with ‘predictors’

Lines 305-06: Replace ‘…(both values expressed on dry matter basis)’ with ‘…,on dry matter basis’

References

Most of the references are not in line with the journal guidelines. Please correct them accordingly. For instance, journal names are to be abbreviated.

Author Response

Dear Reviewer ...

We have carefully reviewed your observations and proceeded to make the changes and corrections requested by you. The modifications are incorporated in the new draft, highlighted with change control and some paragraphs highlighted in yellow to achieve greater visibility

attentively...

Reviewer 2 Report

The paper appears to me a nice, first attempt to evaluate 3 fecal indices as predictors of nutrient intake (DM, P and N) of ewes under extensive grazing conditions. The main concern about the paper is the fact that the regression equations for intake estimation have been obtain from only from 11/12 samples so that they need futher validation. This fact should be cleary stated by Authors. Additional comments are given into the commented pdf copy.

Author Response

Dear Reviewer ...

We have carefully reviewed your observations and proceeded to make the changes and corrections requested by you. The modifications are incorporated in the new draft, highlighted with change control and some paragraphs highlighted in yellow to achieve greater visibility

attentively...

This manuscript is a resubmission of an earlier submission. The following is a list of the peer review reports and author responses from that submission.